# Transition program for adolescents with congenital heart disease in transition to adulthood: protocol for a mixed-method process evaluation study (the STEPSTONES project)

Markus Saarijärvi,[1,2] Lars Wallin,[1,3,4] Philip Moons,[1,2,5] Hanna Gyllensten,[1,6] Ewa-Lena Bratt[1,7]

For numbered affiliations see end of article.

**Correspondence to**
Mr Markus Saarijärvi;
markus.saarijarvi@gu.se

## ABSTRACT

**Introduction** Today, the majority of young persons living with chronic conditions in high-income countries survive into adulthood and will need life-long medical follow-up. Therefore, transition programmes have been developed to facilitate transfer to adult care, and to support self-management and independence during adulthood. The Swedish Transition Effects Project Supporting Teenagers with chrONic mEdical conditionS (STEPSTONES) project aims to evaluate the effectiveness of a person-centred transition programme for empowering adolescents with congenital heart disease in transition to adulthood. To understand how the transition programme causes change and how outcomes are created, process evaluation is imperative to assess implementation, context and mechanisms of impact. This protocol aims to describe the process evaluation of the STEPSTONES transition programme.

**Methods and design** Medical Research Council guidance for process evaluation of complex interventions will be the guiding framework for this mixed-method study. The combination of qualitative and quantitative data will capture different aspects of programme delivery. The sample will consist of participants in the STEPSTONES randomised controlled trial (RCT), persons implementing the programme and healthcare professionals. Quantitative data will consist of protocols and routine monitoring documents from the RCT, data collected from patient registries and sociodemographic data to assess the implementation of the intervention. This data will be analysed with quantitative content analysis, along with descriptive and inferential statistics. Qualitative data will consist of participatory observations, logbooks and interviews with persons implementing the programme, participants and healthcare professionals. Analyses will be performed using qualitative content analysis to investigate mechanism of impact, context and delivery. Quantitative and qualitative data will be integrated in the final stage by using a triangulation protocol according to mixed-method guidelines.

**Ethics and dissemination** The study is approved by the Regional Ethical Review Board in Gothenburg, Sweden. Results will be presented in open access, peer-reviewed journals and at international scientific conferences.

### Strengths and limitations of this study

► This is the first process evaluation study of a transition programme for adolescents with congenital heart disease. Therefore, the study is expected to fill a gap in current knowledge on how transition programmes produce change.

► The use of multiple data sources, both quantitative and qualitative, strengthens the quality of the findings by triangulation between data sources.

► Quantitative data in this study rely heavily on routine monitoring documents from the randomised controlled trial, which may affect the trustworthiness of the process evaluation results.

► Process evaluation data will be reported separately from the results from the effectiveness evaluation.

## INTRODUCTION

Today, the majority of young persons living with chronic conditions (CC) in high-income countries survive into adulthood and need life-long follow-up.[1 2] When going from adolescence to adulthood, these young people will face challenges beyond those normally associated with this period in life.[3 4] Thus, transition programmes have been advocated by several guidelines and consensus statements to ensure continuity of care and support self-management and independence in adulthood.[5–7] The current knowledge emphasises that such programmes should be individually tailored, developmentally appropriate and encompass both the physical and psychosocial needs of the adolescent.[8 9] Furthermore, they should include strategies facilitating both transition as a developmental process, and the transfer of care from a paediatric to an adult context.[5] Transition programmes have been evaluated to some extent during the last decade, showing improvements

in outcomes such as patients' knowledge about their condition, increased self-management, reduced delay in accessing adult care and improved disease-specific indicators.[10–13] However, only four randomised controlled trials (RCT) evaluating transition programmes were included in the recent review by the Cochrane Database of Systematic Reviews, and due to the studies lack of sufficient follow-up and rigorous design, the review concluded that existing research on transition programmes is insufficient in proving hard evidence on effects.[13] Moreover, none of these RCTs conducted process evaluations with the aim to assess how these interventions created change by assessing the implementation process, contextual factors and the underlying mechanisms, that led to the outcomes. More evidence is therefore pivotal in explaining the process of care within transition programmes to provide knowledge on causality and replicability of these interventions.

Transition programmes are complex interventions, comprising numerous interacting ingredients. When the effectiveness of a complex intervention is evaluated, the active ingredients producing the outcome are generally unknown. Process evaluation is therefore indispensable in understanding which these active ingredients are and therefore help explain the mechanism of impact and causal pathways.[14–16] Core elements of process evaluation studies are to investigate how complex interventions (eg, transition programmes) create outcomes, how these outcomes and the change they promote are affected by fidelity and potential moderators on the implementation process and the impact of contextual factors and participants' engagement. Thus, process evaluation is not solely interested in whether the intended intervention produces effect, but also in unpacking 'the black box' of the intervention to investigate how effects are made.[17] In addition to investigating aforementioned aspects, process evaluation studies should also explore if the intervention is evaluated properly with regards to internal and external validity of the RCT, and if the intervention can be sustained in practice.[18] Therefore, the Swedish Transition Effects Project Supporting Teenagers with chrONic mEdical conditionS (STEPSTONES) project was established with the purpose of developing, testing and evaluating a person-centred transition programme for adolescents with congenital heart disease (CHD). This project will provide new evidence on the effects, causality and replicability of transition programmes for adolescents with CCs.

## The transition programme

The study in focus of the process evaluation reported here is the STEPSTONES project, which is a person-centred transition programme for adolescents (age 16–18.5 years) with CHD. The programme was developed from a brief transition intervention previously tested in Belgium.[19] To adjust this intervention to the Swedish context, qualitative and quantitative preparatory studies were performed on the target population.[20–23] Furthermore, stakeholder involvement, literature review and behavioural change

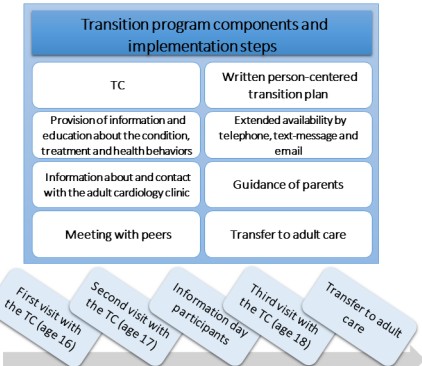

**Figure 1** Key components of Swedish Transition Effects Project Supporting Teenagers with Chronic Medical Conditions transition programme and implementation steps. TC, transition coordinator.

theories were applied, following the protocol for developing health promoting programmes, intervention mapping.[24] The programme aims to be generic, and thus applicable to other CCs. However, CHD was chosen as the target population of this intervention, being the most common congenital malformation in need of life-long follow-up.[25] The programme consists of eight key components, implemented in five steps (see figure 1). A transition coordinator (TC) operating at each intervention centre delivers the programme. The TC is a specialised, paediatric nurse who has received tailored training in delivering the intervention.

## Participants and setting: RCT study

The STEPSTONES programme is evaluated through a hybrid RCT, where a longitudinal observational study is embedded within the RCT (figure 2). The study is performed in seven university hospitals in Sweden where two centres consist of an intervention group (n=70), and a comparison group (n=70) where individual randomisation is performed to either of the groups. The remaining five centres constitute additional control groups (n=70) where no contamination of the intervention is expected. The hypothesis is that adolescents randomised to receive the transition programme over a 2-year period will have

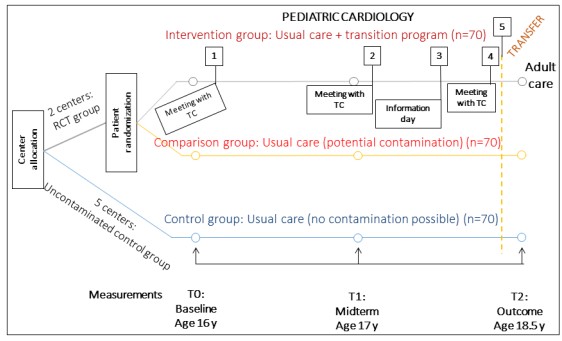

**Figure 2** Overview of the Swedish Transition Effects Project Supporting Teenagers with Chronic Medical Conditions randomised controlled trial (RCT) study design. TC, transition coordinator.

a higher patient empowerment score compared with those randomised to usual care. Inclusion criteria for participants in the RCT study are: 16 years of age with CHD, Swedish speaking and literate. Their parents will also be invited to participate. The study is ongoing and will be concluded in 2021. A study protocol describing the RCT has been published.[26] The process evaluation of the STEPSTONES project is conducted alongside the effectiveness evaluation and health economic evaluation in order to understand and explain the outcomes of the trial. Since the process evaluation is extensive, covering several different aspects of programme delivery, the full extent of this evaluation is reported in this paper.

## Outcome measures of the intervention

The primary outcome of patient empowerment is measured using the Gothenburg Young Persons Empowerment Scale (GYPES). GYPES has been psychometrically tested and proven to be valid for the study population.[27] Patient empowerment is defined as 'an enabling process or outcome arising from communication with the healthcare professional and a mutual sharing of resources over information relating to illness, which enhances the patient's feelings of control, self-efficacy, coping abilities and ability to achieve change over their condition'.[28] The secondary outcomes are transition readiness, disease-related knowledge, health behaviours, patient-reported health, quality of life, healthcare usage and parental uncertainty towards transfer to adult care.[26] The process evaluation is focused on understanding how patient empowerment is achieved within the transition programme and how delivery of the intervention, participants' engagement and contextual factors impact in achieving this outcome.

## Process evaluation of the transition programme

In accordance with the Medical Research Council's (MRC) guidance on the process evaluation of complex interventions, key aspects of this evaluation are centred on the logic model of the intervention, the implementation, mechanism of impact and context.[17] These aspects have a direct impact on how effectively interventions work in producing the desired outcome. The logic model of the intervention provides a structure for designing, carrying out and analysing process data.[17 29] This model can also work as a blueprint of the intervention, describing the underlying theoretical assumptions, input (eg, theory, training and resources), how the intervention is intended to be delivered, components of the intervention and how the intervention aims to cause change.[17] In figure 3, the logic model of the STEPSTONES transition programme is presented along with the key concepts in process evaluation according to the MRC guidance.[17] This figure provides an overview, starting from the logic model, on how the process evaluation of the STEPSTONES transition programme is structured.

The concept of implementation within this model focuses on how successful delivery of the intervention is

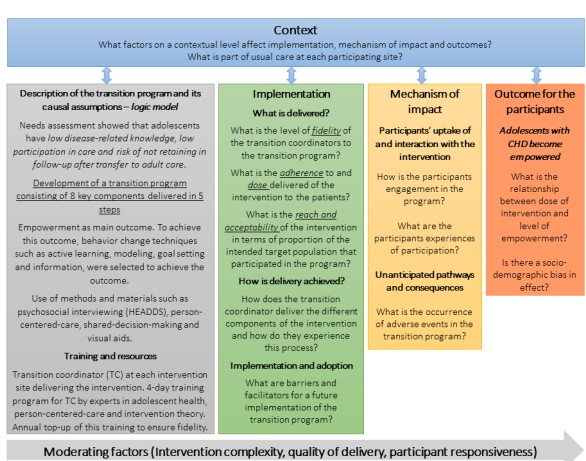

**Figure 3** Overview of the process evaluation of the Swedish Transition Effects Project Supporting Teenagers with Chronic Medical Conditions transition programme, adapted from the Medical Research Council's guidance. CHD, congenital heart disease.

accomplished and what is actually being delivered (ie, fidelity). Fidelity is described as 'the degree to which the clinical intervention was delivered as it was intended' and is assessed by evaluating adherence, dose and reach of each component of the intervention.[30] Fidelity may have a direct impact on outcomes, and thus researchers cannot determine if an intervention failed due to poor implementation or to inadequacies in the programme, unless fidelity is measured.[31] In addition, it is important to assess the process of delivering the intervention. For instance, exploring barriers and facilitators as expressed by the TCs, and how potential moderating factors (eg, quality of delivery) affect how the programme is delivered.[17 30]

The concept of mechanism of impact refers to how the activities performed within the intervention produce intended or unintended effects,[17] in this case the main outcome of patient empowerment. The mechanism of impact for the participants from the intervention will be explored by assessing observable actions of patient empowerment. According to the programme theory, this can include if adolescents have developed the necessary skills to become the manager of their health and care, if they are engaged in the learning process about their health and care and if they remain in follow-up after the transfer to adult care. The mechanism of impact can be studied by including data on participants' responsiveness and interaction with the intervention, their trajectory of care and by assessing possible moderators such as quality of delivery and complexity of the intervention.[17 30]

Finally, context includes factors external to the intervention, which influences implementation, or whether the mechanisms of impact act as intended within the context in which the intervention is being delivered.[17] Contextual factors have shown to impact how the intervention is adapted locally and, in addition, shape the outcomes of the intervention.[32] Therefore, the overall aim is to perform a process evaluation study, assessing

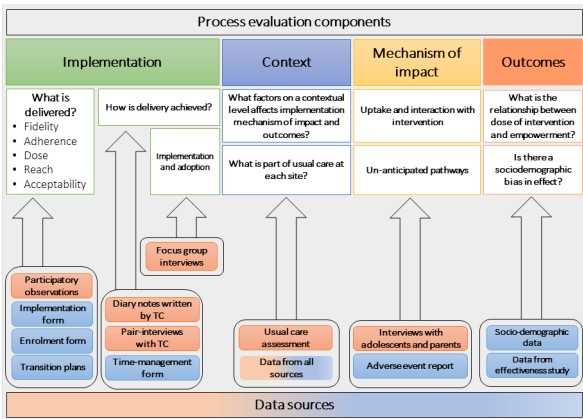

**Figure 4** Overview of Swedish Transition Effects Project Supporting Teenagers with Chronic Medical Conditions process evaluation components and data sources. TC, transition coordinator.

the implementation of the programme and the contextual impact on this process in order to understand and explain the mechanisms creating the outcome of the transition programme. The research questions in relation to the key process evaluation concepts are listed together with the proposed research methods in figure 4.

## METHOD
A mixed-method embedded design will be used, including collection, analysis and integration of quantitative and qualitative data.[33] During the designing and planning of the data collection, the process evaluation guidance by the MRC[14] was chosen as the methodological framework.

### Participants and setting
For the quantitative assessment, participants (n=70) and participating parents in the intervention group will be included. In addition, the TCs at the intervention centres, as well as a sample of physicians and nurses at all participating centres will be asked to participate. For the qualitative assessment, we will use a purposive subsample since it is not considered feasible to include all participants in the intervention group (n=70) in the qualitative data collection. Specifically, we will strive for variability of the participants' characteristics in order to capture a breadth of experiences.[34]

### Data collection
Data will consist of a range of sources in order to capture different perspectives of key process variables. Quantitative and qualitative data collection will be carried out simultaneously and synthesised in the final analysis.

#### Quantitative data collection
The following data sources will be used to collect quantitative data on process measures:
1. Enrolment form
   – The enrolment form is based on the five delivery steps of the intervention (see figure 1). The data al-

low for the measurement of adherence and attendance in the intervention for all eligible participants. Drop out from the study, non-attendance for visits and reasons for this will also be registered, which gives information about acceptability of the transition programme.
2. Intervention implementation form
   – This form will be filled out by the TC after each individual patient encounter, whether in person, by telephone, text message or email. This form contains the components of the intervention, for example, information and education about the health condition, treatment and health behaviour, and information about the adult care follow-up. It also gives information about topics covered, supportive activities and recommendations given, providing an indication about adherence, and dose delivered of the intervention to the patients. The data source will also give information on which communication channel is primarily used for contact between the TC and patients.
3. Adverse event (AE) report
   – The occurrence of AEs will give an indication of potential attrition from the study.
4. Transition plans
   – Each patient in the intervention receives a transition plan in which the care is documented. An analysis of this plan will be undertaken to understand what the main objectives were in the transition of individual patients. This analysis will estimate which components of the intervention have been delivered, the outcomes of these and how this is documented. The data from these sources will give information about the possible mechanism of impact, adherence, dose and the potential effects of moderating factors on fidelity such as patient characteristics.
5. Sociodemographic data
   – Age, sex, education level, diagnosis and pharmacotherapy, comorbidities, geographic location, living situation, country of birth, parental marital status and employment situation will be retrieved from the Swedish registry of CHD (Swedcon) and the questionnaires used in the RCT. These data will be used to examine the characteristics of participants who engage with and continue participation in the intervention, and those who do not (eg, drop outs and non-participants), and how this affects the reach of the programme.
6. Time-management form
   – In this form, the TCs register how much time is spent on performing different tasks of the transition programme, such as the preparation of visits, patient contacts, administration and contacts with other healthcare professionals. Time spent on study-related tasks (eg, recruitment, randomisation and data entry) is excluded from this form. By collecting data on time management, differences in intervention delivery between the two centres will be

visible, also giving valuable information to a future implementation scenario.

## Qualitative data collection

The qualitative data will be used to answer questions on how implementation is achieved, possible moderating factors and how the participants experience and engage with the intervention. Qualitative data will be collected and analysed by a researcher (MS) that is not involved in the RCT study, to ensure trustworthiness of the data.[17] The following qualitative data sources will be used:

1. Pair interviews with TC
   – Throughout the course of the trial, pair interviews[35] with the TCs will be performed every 6 months. An interview guide will be used covering topics such as the experiences of implementing the intervention, barriers and facilitators and local contextual factors.

2. Diary notes written by the TC
   – Reflective diary notes are requested from the TCs, describing their experiences of implementing the transition programme, barriers and facilitators and contextual factors they perceive to have affected the intervention.[36]

3. Participatory observations
   – Observations of outpatient visits will be performed to investigate how the transition programme is being implemented in practice and the responsiveness and engagement of the participants. The observations will provide data on several process variables such as fidelity, adaptations, dose and participants engagement. Both a structured observation protocol and field notes will be used. The protocol includes components from the intervention implementation form and, in addition, practical applications of how the intervention is intended to be delivered, for example, use of materials and methods to deliver the behavioural change techniques and application of person-centered care. Informal interviews will be conducted with the TCs and participants in connection with the observations as a way of ensuring trustworthiness.[37]
   – Observations of an information day for adolescents and their parents will be held between the second and third outpatient visit in the transition programme. During the information day, adolescents and their parents will get the opportunity to meet their peers, receive information about the adult care and discuss issues regarding transfer and transition. Participatory observations of these meetings will be performed, including informal interviews with the participants about their experiences of the day.

4. Semistructured interviews with adolescents and parents
   – Participants and their parents will be interviewed about their experiences 1 month after completion of the transition programme. The interviews will give increased understanding of the dose received by the participants, potential moderating factors on intervention delivery, acceptability of the intervention and potential improvements for a future implementation of the transition programme into current practice.

5. Focus group interviews with the healthcare team
   – Focus group interviews will be held with interdisciplinary teams from the intervention centres. The aim is to explore what contextual factors on a local and organisational level are perceived to affect implementation and what prerequisites that are needed for a future implementation of the transition programme. The sample will consist of physicians and nurses from both paediatric cardiology and adult care with experiences of young persons with CHD.[38]

6. Usual care assessment
   – Throughout the course of the trial, the assessment of the usual care in the control and comparison groups will be undertaken through participatory observations of outpatient visits and interviews with healthcare professionals. The objective is to assess current practices, barriers and facilitators for future implementation and identify possible contamination between study sites.

## Data analysis

### Quantitative data analysis

The process variables (eg, adherence and dose) retrieved from the enrolment form, intervention implementation form and sociodemographic data will be entered into a database and analysed in the Statistical Package for the Social Sciences by using descriptive statistics with frequency distributions, proportions, variability over time and means. The effect on process variables of intervention centres' characteristics and patients' sociodemographic factors will be analysed by regression analysis, as will data from the time management form, to explain potential moderators on fidelity. The reach of the intervention (ie, did the intended population participate in the study) will be assessed by comparing participants versus non-participants of the RCT with sociodemographic data and clinical data (eg, disease characteristics). The MRC guidance recommends reporting results of the process evaluation separately from the effectiveness evaluation to avoid biased interpretation of effectiveness results.[17] We are following this recommendation but will in addition perform a post-hoc regression analysis on dose–outcome relationships as well as subgroup analysis for relevant clinical and demographical variables.

The data from the transition plans will be analysed using descriptive statistics along with quantitative content analysis[39] using the software NVivo (QSR International Pty V.12, 2018) by counting word occurrences related to the components of the intervention to summarise what the main objectives were of each individual participant's transition. AEs will be analysed if they occur.

## Qualitative data analysis

Design, sampling, analysis and reporting of the qualitative findings will follow the Consolidated Criteria for Reporting Qualitative research.[40] Qualitative data will be analysed with manifest content analysis, as described by Graneheim and Lundman.[41] The MRC guidance for process evaluation will act as a coding matrix for the qualitative data in the analysis phase.[17] All qualitative data analysis will be performed in the software NVivo.

## Integrating quantitative and qualitative data

Analysis and reporting of the mixed-method study will follow the Good Reporting of A Mixed Methods Study framework.[42] The mix of quantitative and qualitative data can give information about contextual factors that affect implementation and potential moderating factors on fidelity.[17] Qualitative data can be used to explain quantitative findings.[33] The integration of data will be performed by creating a triangulation protocol, which works as a scheme where the different data sources are combined and findings from different components of the process evaluation are listed and compared. By looking for agreement and disagreement between the data sources, the findings may lead to a better understanding of the results.[43]

## Patient and public involvement

The STEPSTONES project was developed with the support of an advisory board and panel of international experts consisting of stakeholders, that is, young adults with CHD, parents, clinicians, researchers and representatives from the patient organisation. Patients will not be involved in the recruitment of this study; however, young adults with CHD will be involved during the adolescent day sharing their experiences of the transfer to adult care and transition to adulthood, acting as role models and providing peer support. Dissemination of results to study participants will be delivered through the patient organisation and social media.

## Ethics and dissemination

Adolescents, parents and healthcare professionals in the RCT have consented to participation after receiving written and oral information about the study. Written consent to participate in the qualitative data collection will be requested after additional written and oral information about the study. Ethical approval for the study was received from the Regional Ethical Review Board in Gothenburg, Sweden (No. 931-15), and the study will be performed in accordance to the Declaration of Helsinki.[44] Findings of the process evaluation study will be published in open access, peer-reviewed scientific journals and presented at national and international conferences.

## DISCUSSION

Process evaluation studies have been increasing in recent years and are imperative to explaining how complex interventions produce the desired outcomes. Furthermore, such evaluation studies are arguably more important when interventions fail to produce the intended outcomes. Since its release in 2015, MRC guidance has been used in process evaluation studies, providing extensive data on important aspects of interventions delivery.[45 46] However, MRC does lack certain potentially important aspects, for example, the impact of recruitment and the internal and external validity of the RCT on outcomes. For this reason, we have added these research questions to our protocol.

The proposed study has several strengths. First, solid funding of the process evaluation has allowed us to employ adequate number of research staff to handle logistics regarding recruitment, data collection and analysis. Furthermore, the documentation of process variables is digital and can be followed up continuously from the main research site.

Second, when implementing a complex intervention, a clear description of the intervention is beneficial, as a benchmark for evaluators and for those delivering the intervention. Previous studies show that complex interventions with clear descriptions have a greater chance of being implemented with high adherence.[47] The logic model of our intervention was developed using the framework of intervention mapping, which provides a thorough evidence base.[24] In addition, the materials, resources and training for the TCs can facilitate delivery, enhancing intervention fidelity.[30] This process evaluation study can thus add to the existing evidence on how frameworks, such as intervention mapping, can aid in intervention development and evaluation.[24]

Third, the planned extent of data sources, methods and scope of data collection have, to the best of our knowledge, not been applied in previous process evaluation studies. This is beneficial for many reasons. The use of various sources can strengthen understanding of how different factors interact in creating outcomes. A recent review concluded that only one quarter of all studies measuring fidelity and engagement in interventions used multiple data collection methods such as observations and self-reported measures.[48] A range of qualitative methods within intervention studies can therefore strengthen the evaluation since these methods are forceful in capturing contextual factors.[32 49] Moreover, quantitative data provide measures for important process variables such as fidelity.[30] By collecting data throughout the trial, this study can also handle the impact of 'teething problems' (eg, TCs learning curves) by following changes over time and combining quantitative measures with stakeholder interviews.[17]

Finally, the findings of the process evaluation will be reported separately from the effectiveness evaluation, avoiding biased interpretations of results and therefore strengthening the validity of the findings.[17] Since the process measures in this study rely on routine monitoring data, validity of these documents has to be ensured by quality control and cross checking between different

sources. Routine monitoring data are effective in reducing response bias; they are cost effective and allow for analysis over time.[17]

However, some challenges and limitations of this process evaluation study have to be considered. Factors affecting recruitment are not directly included in the qualitative data collection. In Steckler and Linnan's[50] framework, this is a part of the process evaluation which is not included in the MRC guidance.[17] Recruitment pace, selection of study participants and recruiters' skills could arguably be considered moderating factors on reach and outcomes. Moreover, the considerable amount of questionnaires distributed to participants in the effectiveness evaluation might affect willingness to participate and therefore adherence. That said, the acceptability of the intervention is explored and measured, and so is reach in terms of participants approached and recruited. Furthermore, the TCs performing the intervention, and their performance during the intervention, will have a great impact on many of the key process measures since they are the only personnel delivering the intervention. To manage this, data will also be collected from participants and healthcare professionals in order to broaden the perspective. Process evaluators should also be aware of how they impact the quality of the collected data (eg, Hawthorne effects), especially in participatory observations.[51] However, a small subsample of qualitative observations from a total of 210 outpatient visits should arguably give little or no impact on effectiveness outcomes of the RCT. Nevertheless, it may affect the quality of the process evaluation results. To manage this, establishing good relations between the researchers performing the process evaluation and the TCs and healthcare professionals is important to get access to the field and retrieve trustworthy data. Informal interviews in connection with observations can also strengthen trustworthiness.[49]

## CONCLUSIONS

Transfer of care for adolescents living with CCs is a timely issue, and the effectiveness and causal assumptions of transition programmes are yet to be proven. It is therefore imperative to conduct high-quality process evaluation studies to understand how the programmes are delivered, for whom they are delivered and under what circumstances. The effect of contextual factors and possible moderating factors on intervention fidelity and participant engagement may also increase transferability of findings and can thus inform the implementation of transition programmes in other settings.

**Author affiliations**
[1]Institute of Health and Care Sciences, Goteborgs Universitet, Goteborg, Sweden
[2]Department of Public Health and Primary Care, KU Leuven, Leuven, Vlaanderen, Belgium
[3]School of Education, Health and Social Studies, Hogskolan Dalarna, Falun, Dalarna, Sweden
[4]Department of Neurobiology, Care Sciences and Society, Karolinska Institute, Stockholm, Sweden
[5]Department of Paediatrics and Child Health, University of Cape Town, Cape Town, South Africa
[6]Centre for Person-Centered Care, Goteborgs Universitet, Goteborg, Sweden
[7]Department of Pediatric Cardiology, Drottning Silvias barn- och ungdomssjukhus i Goteborg, Goteborg, Sweden

**Acknowledgements** The authors would like to acknowledge and thank the members of the panel of international experts and of the Advisory Board.

**Contributors** MS, LW, PM,HG and ELB conceived the study and were responsible for the design and draft of the study protocol. All authors critically revised and approved the manuscript.

**Funding** This work was supported by research grants from the Swedish Heart-Lung Foundation (grant 20150535); Swedish Research Council for Health, Working Life and Welfare-FORTE (grant STYA-2015/0003); Swedish Children Heart Association; Swedish Research Council (grant 2015-02503); and the Institute of Health and Care Sciences of the University of Gothenburg.

**Competing interests** None declared.

**Patient consent for publication** Not required.

**Ethics approval** The study has received ethical approval from Regional Ethical Review Board in Gothenburg, Sweden.

**Provenance and peer review** Not commissioned; externally peer reviewed.

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
