## [Reviewer comments · BMJ Open]

ARTICLE DETAILS

TITLE (PROVISIONAL)	Transition program for adolescents with congenital heart disease in transition to adulthood: protocol for a mixed-methods process evaluation study (the STEPSTONES project)
AUTHORS	Saarijärvi, Markus; Wallin, Lars; Moons, Philip; Gyllensten, Hanna; Bratt, Ewa-Lena

VERSION 1 – REVIEW

REVIEWER	Ladouceur, Magalie Hôpital Européen Georges Pompidou, France
REVIEW RETURNED	16-Dec-2018

GENERAL COMMENTS	The authors report a protocol for process evaluation of the person-centered transition program for patients with congenital heart disease, the STEPSTONES program. Participants in the STEPSTONES randomized controlled trial, as well as persons implementing the program, and healthcare professionals will constitute the sample of the study. They used the Medical Research Council guidance for Process evaluation of complex interventions combining qualitative and quantitative data. Several points underlight the quality of this research: 1/ Process evaluation is imperative to understand how complex interventions, such as a transition program for adolescents with CHD, produce the desired outcomes. The outcomes of a such study will improve knowledge on how transition programs produce effect in this population.2/The integration of qualitative data and quantitative data, using a triangulation model, will improve quality of findings.3/ Finally, process evaluation is performed separately from effectiveness evaluation, to avoid biased interpretation of effectiveness results. The design of the study is very well developed in the Methods section. The only minor comment concerns the dates of the study which should be included, even if they may correspond with the STEPSTONES RCT which is detailed in the reference 26.
--

REVIEWER	Johanna Calderon, PhD, Assistant Professor of Psychiatry Harvard Medical School, Boston Children's Hospital
REVIEW RETURNED	15-Jan-2019

GENERAL COMMENTS	The authors propose to conduct an important study on the process evaluation of a complex intervention program STEPSTONES transition program for adolescents with Congenital Heart Disease. The protocol of this randomized controlled trial was previously
---

	published. Adolescents and young adults with CHD constitute a growing patient population in need of transition and clinical services. This study is germane in the analysis of the mechanisms underlying a transition program for adolescent empowerment. The authors propose to carry out an important work. However, the way the study is presented lacks significant clarity as to what the major outcomes will be and why this study (or at least part of it) was not embedded in the original randomized controlled trial. A few comments on this paper:  - The key process variables may be more explicitly presented. - The Research questions should be included in the article to help the reviewer understand the specifics of this design. As it is, it is on Figure 4 only. - The quality of the figures are not optimal. It is difficult to read some parts of them. - Please provide further information regarding why you will only use a sub-sample for the qualitative methodological part of the study? What are the methodological guidelines mentioned? -Please review the English of the paper (e.g., enrollment; data used with plural, etc). The manuscript is quite dense and lacks some clarity as to what the authors are trying to achieve.
--	---

REVIEWER	Sumeet Vaikunth Lucile Packard Children's Hospital Stanford, USA
REVIEW RETURNED	18-Feb-2019

GENERAL COMMENTS	This protocol study focuses on the important topic of transitional care in congenital heart disease and is unique for its emphasis on process evaluation. I look forward to its results and implications for what specific aspects of the process improve patient empowerment as well as the secondary outcomes.
--

REVIEWER	Dimopoulos, Konstantinos Royal Brompton Hospital, Adult Congenital Heart Centre and Centre for Pulmonary Hypertension
REVIEW RETURNED	20-Feb-2019

GENERAL COMMENTS	This is a very well written paper present a complex protocol on a groundbreaking study on transition of patients with congenital heart disease. A study on this topic is urgently required and the authors appear clearly to be experts in the field. I have the following comments: How does the current protocol differ to the published protocol in the same journal by Acuña Mora, et al. Person-centred transition programme to empower adolescents with congenital heart disease in the transition to adulthood: a study protocol for a hybrid randomised controlled trial (STEPSTONES project). BMJ Open. 2017 Apr 17;7(4):e014593. It appears to be the same study. If it is a different study how did the 2017 version of the study conclude and are there any pilot data to be used? I have major concerns with regards to the feasibility of this study. It is extremely ambitious, with an intervention spanning over 2 years, and a very high number of variables collected, including several questionnaires to individual patients. One would have concerns that patients answering dozens of questions may lose concentration and the quality of information collected may
---

	deteriorate as the number of questionnaires increases. Can the authors explain how they will overcome these logistic issues. Have the questionnaires used been validated? have the authors an understanding of what is the range of expected scores and what is clinically and statistically significant? Can data from previous studies using the same questionnaires be used to calculate the sample size and perform power calculations. Have the authors considered a performing a pilot study to assess feasibility, and help power the main study appropriately? Transition in many countries is said to start at the age of 12 years can the authors explain why these studies starting at 16?
--	--

VERSION 1 – AUTHOR RESPONSE

Reviewer 1	
Comment	Answer
The only minor comment concerns the dates of the study which should be included, even if they may correspond with the STEPSTONES RCT which is detailed in the reference 26.	Dear Dr Ladouceur Thank you for the valuable comments regarding our manuscript. We have added the dates to when the study is performed on page 5 before reference 26.
Reviewer 2	
However, the way the study is presented lacks significant clarity as to what the major outcomes will be and why this study (or at least part of it) was not embedded in the original randomized controlled trial. The key process variables may be more explicitly presented.	Dear Dr Calderon Thank you for the constructive feedback and thorough review of our manuscript. These comments provided us with valuable input on how to improve this paper and study as a whole. We have worked with improving the clarity of the manuscript by emphasizing the process evaluation earlier on in the introduction and how this is linked to the primary outcome. Furthermore, the key process variables and the link to the outcome of patient empowerment is described in figure 3 and 4 which we have worked in improving in regards to clarity. Regarding your concern on why this study was not embedded in the original RCT we will try to answer it below. First, when developing the intervention we followed the MRC Framework for development and Evaluation of Complex Interventions (Craig, et al 2008, 2013) which emphasizes that complex interventions needs to be evaluated in regards to

	effectiveness, cost-effectiveness and process of care. The process evaluation is not embedded in the RCT's effectiveness evaluation, however it is added as a separate evaluation in order to understand and explain the outcomes of the trial. Second, the process evaluation along with the health economic evaluation of the RCT received funding after funding was received for the effectiveness evaluation. Third, since receiving funding for these studies separately we decided to design a thorough process evaluation. This because transition programs are complex interventions and their causal mechanisms are unknown. Therefore, an extensive process evaluation can answer this vital question. Fourth, since the process evaluation study is extensive and involves different data sources and methods we decided to write this study protocol separately from the protocol describing the effectiveness evaluation to link all the parts of the process evaluation together in a comprehensive way. Fifth and finally, we plan to publish several studies on the process evaluation according to the MRC guidance (Moore et al 2015), due to the extensive data collection. Therefore, this protocol paper can act as a matrix, linking together the different parts of the process evaluation.
The Research questions should be included in the article to help the reviewer understand the specifics of this design. As it is, it is on Figure 4 only.	We have specified the aim in the article by adding information on what aspects the process evaluation will investigate. However, since we have many research objectives in relation to each method and concept we are using the figure as a visual representation of how the different components of the process evaluation is linked to the research questions and related data sources. In order to make it easier for the reader to assess the design we have increased the font size and resolution of the figure.
The quality of the figures are not optimal. It is difficult to read some parts of them.	Thank you. We have increased the font size and highlighted some aspects of the figures in bold to increase the readability.
Please provide further information regarding why you will only use a sub-sample for the qualitative	The sub-sample for the qualitative data collection is used for the interviews with patients and parents and for the participatory observations.

methodological part of the study? What are the methodological guidelines mentioned?	This, because it is not feasible to interview or observe all patients included in the study since 70 patients are participating. The methodological guidelines are written by Tong et al. and is reference number 34 in this section. These guidelines are widely used in qualitative research.
Please review the English of the paper (e.g., enrollment; data used with plural, etc).	Thank you for noticing. The manuscript has been copy-edited by a native English-speaking translator.
Reviewer 3	Dear Dr Vaikunth Thank you for the positive comments about our study.
Reviewer 4	Dear Dr Dimopoulos Thank you for the comments and feedback about our study. Below we will answer to your questions raised.
How does the current protocol differ to the published protocol in the same journal by Acuña Mora, et al. Person-centred transition programme to empower adolescents with congenital heart disease in the transition to adulthood: a study protocol for a hybrid randomised controlled trial (STEPSTONES project). BMJ Open. 2017 Apr 17;7(4):e014593. It appears to be the same study. If it is a different study how did the 2017 version of the study conclude and are there any pilot data to be used?	The protocol published by Acuña Mora, et al described the effectiveness evaluation of the RCT. Since the transition program is a complex intervention, the RCT should also be evaluated in terms of process and health economic consequences. Since the present study is an extensive process evaluation study, imperative to answer the question on how transition programs cause change, we decided to write a separate protocol paper describing the process evaluation as a whole. These studies have become increasingly common within research on complex interventions and protocols describing process evaluation alongside randomized controlled trials have been published previously in BMJ Open. The process evaluation is performed on the transition program being evaluated in the RCT described by Acuña Mora, et al. The study is ongoing and outcome data will be available in 2021.
I have major concerns with regards to the feasibility of this study. It is extremely ambitious, with an intervention spanning over 2 years, and a very high number of variables collected, including several questionnaires to individual patients. One would have concerns that patients answering dozens of questions may lose concentration and the quality of information collected may deteriorate as the number of questionnaires increases. Can the authors	Yes, this is an ambitious study with a long follow-up. However, in 2016 the Cochrane Database of systematic reviews concluded that hard evidence on effectiveness of transition programs for adolescents with chronic conditions is lacking. One reason for this is that previous studies have had a too short follow up period of 6 months to 1 year. In order to present long-term outcomes and

explain how they will overcome these logistic issues.	to understand how these interventions cause change there is a need of transition programs that have a follow up of more than 2 years. Regarding your question of feasibility and logistics of the RCT, we will answer your concern below. First, solid funding of this RCT has given us the prerequisites to employ adequate number of research staff to handle logistics regarding recruitment, data collection and analysis. This both centrally from the main site from where the study is monitored and at each enrolled hospital. Second, the project coordinator, principal investigator and research group are continuously monitoring the project and following up on the logistics. Third, all documentation in the trial (recruitment, data collection, enrolment and questionnaires) is digital and can be followed up continuously from the main research site. Fourth, frequent site visits and monitoring several times a year ensures fidelity. Furthermore, process evaluation studies (such as this) can in addition to answer questions on possible mechanisms leading to the outcomes, also investigate fidelity and adherence, which this study is undertaking. One aspect of fidelity is patient adherence that is moderated by potential dropouts. The present effectiveness study has been ongoing for the last 2 years and we have not yet experienced issues with patients not filling out the questionnaires. However, if this would be an issue we will investigate this further in the interviews with the adolescents after participation in the study, which will be performed as a part of this process evaluation. We thank you for raising this concern.
Have the questionnaires used been validated? have the authors an understanding of what is the range of expected scores and what is clinically and statistically significant?	Yes, the questionnaires have been validated. The questionnaire measuring the primary outcomes of patient empowerment is referred to in this manuscript (no 27) and has been previously published. Sample size calculation is based on the primary outcome of empowerment. The target is an improved patient empowerment score of 5.25 points on a scale from 15 to 75 (ie, 0.5 SD).

	For two-sided tests with $\alpha=0.05$ and power=80%, 63 patients are needed in each arm of the RCT. In order to compensate for a potential 10% dropout rate, we will recruit 70 patients in each arm of the RCT. An additional 70 patients will be recruited in the centres of the control group. Among the seven centres, a total of 210 patients will be enrolled (27).
Can data from previous studies using the same questionnaires be used to calculate the sample size and perform power calculations. Have the authors considered a performing a pilot study to assess feasibility, and help power the main study appropriately?	Sample size and power calculation of this has been described in the paper published by Acuña Mora, et al. (no 26 and 27)
Transition in many countries is said to start at the age of 12 years can the authors explain why these studies starting at 16?	Yes, this is correct and we are well aware of that transition preparation should start earlier according to current guidelines. However, for this study, we received research funding for a period of 3-4 years and therefore we cannot deliver or implement a transition program within a RCT that is longer than 2 years in order to have time for recruitment and analysis within this timeframe.

VERSION 2 – REVIEW

REVIEWER	Sumeet Vaikunth Lucile Packard Children's Hospital, Stanford, USA
REVIEW RETURNED	11-May-2019
GENERAL COMMENTS	I commend the authors for their thoughtful responses to the reviewers. I believe this study does have the major strength, as the authors mention, of being the first study focused on process improvement in transition of care and is therefore worthy of publication in its current form. The funding support and research personnel support as described by the authors should allow for completion of this study. I look forward to its results.
REVIEWER	Dimopoulos, Konstantinos Royal Brompton Hospital, Adult Congenital Heart Centre and Centre for Pulmonary Hypertension
REVIEW RETURNED	07-May-2019
GENERAL COMMENTS	The authors have answered all reviewer comments appropriately, even though they have made no changes/additions/clarifications to their paper in response to many of the comments.

VERSION 2 – AUTHOR RESPONSE

Reviewer 4 – Konstantinos Dimopoulos -The authors have answered all reviewer comments appropriately, even though they have made no changes/additions/clarifications to their paper in response to many of the comments.	Dear Dr Dimopolous, Thank you for appraising our responses to the comments. We have now revised the manuscript to further address the comments you and your fellow reviewers have provided to us. Please see table 2, where we have described our revisions in the text.
Reviewer 3 - Sumeet Vaikunth I commend the authors for their thoughtful responses to the reviewers. I believe this study does have the major strength, as the authors mention, of being the first study focused on process improvement in transition of care and is therefore worthy of publication in its current form. The funding support and research personnel support as described by the authors should allow for completion of this study. I look forward to its results.	Dear Dr Vaikunth, Thank you for appraising our responses to your comments.